# Silent Death by Sound: C_60_ Fullerene Sonodynamic Treatment of Cancer Cells

**DOI:** 10.3390/ijms24021020

**Published:** 2023-01-05

**Authors:** Aleksandar Radivoievych, Benjamin Kolp, Sergii Grebinyk, Svitlana Prylutska, Uwe Ritter, Oliver Zolk, Jörn Glökler, Marcus Frohme, Anna Grebinyk

**Affiliations:** 1Division Molecular Biotechnology and Functional Genomics, Technical University of Applied Sciences Wildau, Hochschulring 1, 15745 Wildau, Germany; 2Department of Chemistry, Taras Shevchenko National University of Kyiv, Volodymyrska 64, 01601 Kyiv, Ukraine; 3Institute of Chemistry and Biotechnology, Ilmenau University of Technology, Weimarer Straße 25 (Curiebau), 98693 Ilmenau, Germany; 4Institute of Clinical Pharmacology, Brandenburg Medical School, Seebad 82/83, 15562 Rüdersdorf, Germany

**Keywords:** ultrasound, C_60_ fullerene, sonodynamic therapy, HeLa cells, apoptosis

## Abstract

The acoustic pressure waves of ultrasound (US) not only penetrate biological tissues deeper than light, but they also generate light emission, termed sonoluminescence. This promoted the idea of its use as an alternative energy source for photosensitizer excitation. Pristine C_60_ fullerene (C_60_), an excellent photosensitizer, was explored in the frame of cancer sonodynamic therapy (SDT). For that purpose, we analyzed C_60_ effects on human cervix carcinoma HeLa cells in combination with a low-intensity US treatment. The time-dependent accumulation of C_60_ in HeLa cells reached its maximum at 24 h (800 ± 66 ng/10^6^ cells). Half of extranuclear C_60_ is localized within mitochondria. The efficiency of the C_60_ nanostructure’s sonoexcitation with 1 MHz US was tested with cell-based assays. A significant proapoptotic sonotoxic effect of C_60_ was found for HeLa cells. C_60_′s ability to induce apoptosis of carcinoma cells after sonoexcitation with US provides a promising novel approach for cancer treatment.

## 1. Introduction

Closed-sphere carbon nanostructure C_60_ fullerene [1] (here consistently abbreviated “C_60_”) is used for several biomedical applications since its unique structure can elicit antiviral, antimicrobic and anticancer activities [2]. The specific packing of sixty carbon atoms in penta- and hexagon units arrange a rather unusual sp^2,3^ hybridization structure [3] with a surface three times smaller than expected for its respective molecular weight. The pristine C_60_ has very low solubility in water. Derivatization and colloid solutions are used to increase C_60′_s solubility in aqueous solutions, which is critical for biological application [4]. Functionalization of C_60_ improves its water solubility and increases its biocompatibility by decreasing the aggregate size [5], but on the other hand, it inhibits its interaction with cellular lipid membranes and changes the pattern of cellular uptake [5,6,7,8,9]. The pristine C_60_ can form aggregates in aqueous solutions and make stable colloid solutions that contain both individual C_60_ and its nanoclusters [10,11,12]. Small-angle neutron scattering (SANS) and atomic force microscopy (AFM) evidenced that the C_60_ aqueous colloidal solution remained stable for six months. The value of Zeta potential for pristine C_60_ aqueous colloid solution was found to be on the level of from −30 to −10 with a maximum of −23 mV [13]. Given carbon bonds similar to the planar graphene, C_60_′s non-planar π-conjugated system of molecular orbitals determines its significant absorption of UV-VIS light. The UV-VIS absorption spectrum of pristine C_60_ fullerene aqueous colloidal solution has three intense absorption bands typical for C_60_ with maxima around 215, 265, and 350 nm and a long broad tail up to the red region of the visible light [13,14]. After light absorbance, a photoexcited C_60_ molecule can generate reactive oxygen species (ROS) through energy or electron transfer to oxygen [15]. The low photobleaching, high quantum yield and photostability [16] of the C_60_ molecule boosted the rapid development of its application in cancer therapy as a photosensitizer [2,15,16,17]. Previously, negligible toxicity of pristine C_60_ [18] and its colloid solution [11,19,20] against normal cells was shown. Considerable concentrations (277 µM) had no effect on morphology, cytoskeletal organization, cell cycle dynamics, or the proliferation of normal human mammary epithelial MCF10A cells [18]. Prylutska et al. [20] and Tolkachov et al. [21] proved that C_60_ aqueous colloidal solution, explored in the current study, was nontoxic at low therapeutic doses for normal models in both in vitro and in vivo systems. Thus, C_60_ aqueous colloid solution at concentrations from 6 to 24 µg/mL (8–33 µM) did not manifest in vitro toxic effects on normal cells such as thymocytes, macrophages and hepatocytes [21]. C_60_ aqueous colloidal solution also demonstrated low toxicity against human embryonic kidney cells with a high IC_50_ value (555 µM at 24 h) [20]. Moreover, no effect of C_60_ aqueous colloidal solution in the low doses (75 and 150 mg/kg) on the behavioral reaction of mice was detected. The LD_50_ value for C_60_ fullerene was 721 mg/kg [20].

At the same time, a pronounced ROS-mediated proapoptotic effect through a mitochondrial pathway was detected in cancer cells treated with pristine C_60_ and irradiated with visible light [14,22,23]. The light irradiation (λ = 320–580 nm) of the C_60_ at 10^−6^, 10^−5^ and 10^−4^ M resulted in the rate of ROS production on the level of 1.1 ± 0.9, 3.4 ± 0.2 and 10.4 ± 1.7 nmol/min accordingly [23]. A pronounced ROS-dependent proapoptotic effect was detected in leukemic cells treated with ≤20 µM C_60_ and irradiated with UV-VIS light in the same range of 320–580 nm [4,10,12,23,24,25]. A continuous intensification of ROS production and inhibition of the glutathione-dependent antioxidant system testified to the subsequent intense induction of oxidative stress [24]. The further data proved C_60′_s ability to induce ROS production and apoptosis of leukemic cells after photoexcitation with high power single chip 405 nm LED [14]. The further development of C_60_ as a photosensitizer in the frame of cancer photodynamic therapy (PDT) is hampered by its relatively high band gap [26] and low absorbance of the tissue penetrating long-wavelength [27] light. Moreover, PDT faces the heterogeneous nature of biological tissues that can affect the original path of photons due to the high absorption, scattering and anisotropy [28].

The deep penetration of ultrasound (US) waves in biological tissues beyond the reach of external light has promoted the idea of using them as an alternative energy source for the excitation of photosensitizers. Sonodynamic therapy (SDT), derived from the PDT, recently emerged as a non-invasive cancer treatment modality relying on the activation of certain chemical sensitizers with US. It has been generally accepted that the cavitation effect of US is responsible for the SDT mechanism [29]. Acoustic cavitation is a unique physical phenomenon involving the formation, growth and collapse of bubbles during the propagation of US waves in liquids. The explosion of bubbles leads to sonoluminescence that releases the accumulated energy [30]. The sonoluminescence spectrum in water is relatively broadband, with a UV maximum and a long-wavelength tail [31,32,33]. It has been shown that the cavitation bubbles generated by ultrasound not only transform sound into light but also cause pyrolysis and increase the temperature, which can be attributed to the modulation of toxic effects as well [34]. Various organic sonosensitizers have been adopted from PDT to SDT, including aminolevulinic acid [35,36], Rose Bengal [37] and porphyrins [38]. Compared to organic sonosensitizers, inorganic nanoparticles such as gold [39], silicon [40] and titanium dioxide [41] offer relatively high chemical and physiological stability and have also been demonstrated to be effective in SDT. The polyethylene glycol- [42], polyhydroxy [43], tris-acid [44] fullerenes and C_60_/PMPC (poly(2-methacryloyloxyethyl phosphorylcholine)) complexes [45] have also been shown to efficiently induce ROS-mediated compact apoptotic cancer cell death once used in SDT. The pristine C_60′_s higher lipophilicity over its derivatives promotes its faster diffusion across the plasma membrane and facilitates intracellular uptake [7,46]. Owing to the nature of sonoluminiscence and its spectrum in particular [31,32,33], US seems to be a good matching option for activating pristine C_60_ to generate ROS. Herein, we broaden the biological application of C_60_ aqueous colloidal solution with the first data to our knowledge on the use of 1 MHz US for sonosexcitation of C_60_ to treat carcinoma and normal cells, examining the intracellular accumulation of C_60_ and the mechanism of cell death.

## 2. Results

### 2.1. C_60_ Aqueous Colloid Solution

In order to check the most abundant molecular ions in the aqueous C_60_ solution used, the MALDI-TOF-MS (matrix-assisted laser desorption ionization-time of flight mass spectrometry) method was employed. This method can be used to ensure that the preparation of C_60_ in water has not caused any changes to the fullerene structure. The MALDI-TOF-MS analysis of C_60_ samples revealed sharply defined peaks for a predominant molecular mass of 720 Da (Figure 1a).

The obtained spectrum confirms the presence of naturally occurring stable isotopes of the common element carbon, resulting in the gradual triplication of the peak. Only 98.89% of naturally occurring carbon atoms are in the form of ^12^C; most of the remaining 1.11% consists of atoms of ^13^C and a trace amount of ^14^C [47]. The presence of one ^13^C atom in a C_60_ molecule shifted the molecular mass of C_60_ to 721 Da whereas a C_60_ molecule with a molecular mass of 722 Da had two ^13^C atoms in its cage. Alternatively, those peaks could correspond to molecular adduct ions of [M + H] and [M + 2H] in the matrix or in the presence of Trifluoroacetic acid.

In order to check the stability of the aqueous colloid solution, the size distribution was monitored with dynamic light scattering. The average of C_60_ nanoparticles was evaluated to be 120 nm (Figure 1b) which matched previous investigations [13,48] and evidenced storage stability over a period of six months.

### 2.2. Sonoluminescence Detection

In order to prove the existence of sonoluminescence, the optical measurements were done with a sensitive photomultiplier tube, able to detect even a single photon via photoelectric effect and secondary emission. Obtained Vpp (peak-to-peak voltage) data on light intensity were recorded in a US bath during 100–500 W output power of US generator and normalized with the respective Vpp obtained when the shutter of the photomultiplier window was closed with the US on. The detected increase of the Vpp proved the existence of the sonoluminescence during 1 MHz US propagation through degassed distilled water in the water bath (Figure 2). In addition, the level of the detected sonoluminescence was increased with the higher output power of the US generator. Thus, the increase of the output power of the US generator from 100 to 500 W resulted in a 44% increase in the detected Vpp signal. Next, the sonoluminescence level in the water bath with a well plate (microtiter-plate) was investigated to check the sonoluminescence in the well. The obtained Vpp from the photomultiplier tube evidenced the decrease of the sonoluminescence level in the well plate as compared to the sonoluminescence level in the bath. However, the Student’s *t*-test confirmed its dose-dependent significant positive correlation with the output power of the US generator as well. Therefore, it can be concluded that sonoluminescence occurred during 1 MHz US propagation in the experimental set-up required for cell-based assays.

### 2.3. Intracellular C_60_ Accumulation

The intracellular uptake and distribution of C_60_ were studied by fluorescence immunostaining of HeLa cells using a FITC-labeled sandwich of antibodies against C_60_. Figure 3a presents the images of HeLa cells stained after incubation with 20 µM C_60_ for 24 h. Simultaneously, cells were stained with DNA-binding dye DAPI for cell nucleus and membrane-potential-sensitive MitoTracker Orange for mitochondria visualization. The detected green fluorescence evidenced C_60_ uptake and clear extranuclear localization.

To study the accumulation dynamics, we extracted C_60_ from the cell homogenate as well as from the mitochondrial fraction and carried out HPLC-ESI-MS (high-performance liquid chromatography/electrospray ionization tandem mass spectrometry) analysis. The observed time-dependent intracellular uptake of C_60_ reached a maximum of 800 ± 66 ng/10^6^ cells after 24 h of incubation (Figure 3b). The intracellular amount of C_60_ in the HeLa cells was found to be three times higher as compared with human leukemic CCRF-CEM cells, as investigated before [14,49], potentially a result of the much higher cytosol/nucleus volume ratio.

The next step was to quantify C_60_ content in the mitochondria using both fluorescence image processing and HPLC-ESI-MS. C_60_ content in the mitochondria fraction showed accumulation at a level of < 380 ± 30 ng/10^6^ cells at 24 h, representing 47% of its overall cellular content. The yellow color in the merged fluorescence images verified a partial co-localization of green C_60_ antibodies and red mitochondrial marker.

### 2.4. Cell Viability

In order to assess whether sonodynamic treatment of HeLa and HEL 299 cells incubated with C_60_ could have any toxic effects, cell viability was analyzed. Cells were treated with 20 μM C_60_ for 24 h, exposed to 1 MHz US and after a further 48 h, their viability was estimated by MTT (3-(4,5-dimethylthiazol-2-yl)-2,5-diphenyl tetrazolium bromide) assay. The described conditions were selected after “try-and-fail” rigorous comparisons of the US treatment’s effects on the temperature of the liquids in well-plates as well as on cell viability (data not shown). It was found that the US intensity of 5.4 W/cm^2^ could be safely used for the treatment mode for up to 60 s, keeping the temperature under 38 °C without any significant changes in the viability of the cells. The “solvent” control cells, incubated with an equal volume of sterile water and treated with US, were found to exhibit no significant viability changes. The viability of the respective control cells with neither C_60_ nor US treatment was considered 100%. However, the application of US in the presence of C_60_ led to a gradual decrease in the cells’ viability. The US dose of 60 s in the presence of 20 µM C_60_ decreased the cell viability to 59 ± 5% (Figure 4a). Visual changes in cell quantity and morphology were also observed with phase-contrast microscopy. As shown in Figure 4b, HeLa cells, exposed to combined treatment of 1 MHz US and 20 µM C_60_, demonstrated a decrease in viable cells. Our results evidenced that 1 MHz US induced significant cytotoxic effects of C_60_ against human carcinoma cells.

Additionally, viability evaluation of human embryo lung HEL 299 cells was performed to test the proposed treatment modality on a normal healthy cell model. The viability of the respective control cells with neither C_60_ nor US treatment was considered 100%. C_60_ and US treatment alone had no effect on HEL 299 cell viability. The treatment of HEL 299 cells with US in the presence of C_60_ led to a slight decrease in the cells’ viability to 88 ± 5%. Student’s *t*-test showed that this decrement is insignificant in comparison with the viability of cells treated with the respective duration of US (Figure 5).

The results of the cell viability tests allow us to conclude that the sonotoxic effects of C_60_ have selective toxicity against cancer cells, whereas the effect on normal cells was negligible. Based on this, SDT with C_60_ offers the selective induction of cancer cells viability decrease.

### 2.5. Apoptosis Induction

Cytotoxic effects of photoexcited C_60_ are considered to induce the mitochondrial apoptotic pathway of cell death [15,23]. Photoexcited C_60_ generates ROS that leads to the release of cytochrome *c* from mitochondria and the induction of apoptosis through the mitochondrial pathway [22]. Cytochrome *c* initiates apoptosome formation that activates caspase 9 and 3/7 [50]. Later during apoptotic death propagation, cells expose phosphatidylserine (PS) as an ‘eat me’ signal for phagocytes. In normal cells, PS is placed in the inner leaflet of the lipid bilayer, but when cells undergo apoptosis, caspases inactivate flippase, which translocates PS in the inner leaflet of the lipid bilayer, leading to irreversible PS exposure in the outer leaflet of the lipid bilayer [51]. These two phenomena are specific for apoptotic cell death and can be used as markers of apoptosis. A similar sonosensitizing toxicity of intracellular accumulated C_60_-inducing apoptosis had to be proven. Thus, our final goal was to evaluate caspase 3/7 activity and plasma membrane phosphatidylserine translocation evidencing for apoptosis.

No significant effect of either C_60_ or 1 MHz US alone on caspase 3/7 activity of HeLa cells was observed following 3 h of cell incubation. However, treatment of cells with C_60_ and US was followed by an increase in caspase 3/7 activity. Thus, caspase 3/7 activity was increased to 128 ± 13, 162 ± 16 and 342 ± 29% in HeLa cells, incubated with 20 µM C_60_ for 24 h and subjected to 1 MHz US treatment for 20, 40 and 60 s, correspondingly (Figure 6a).

HeLa cells, treated with C_60_ and US, were subjected to double staining with phosphatidylserine-binding Annexin V-FITC and DNA-binding dye propidium iodide (Figure 6b). The control cells had a viability of 88 ± 4%. Neither treatment with 20 µM C_60_ nor US alone had a significant effect on cell distribution profiles, demonstrating a viability rate of 83–87 ± 3%. However, under the combined action of C_60_ and 1 MHz US, a significant increase in the content of apoptotic HeLa cells was detected, which reached the level of 83 ± 4%, compared to 11 ± 1% of control cells, treated with C_60_ and kept in the dark (Figure 6b,c). The obtained data allow concluding that the toxic effect of C_60_ fullerene against HeLa cells after sonoexcitation is realized by apoptosis induction.

## 3. Discussion

The common trend in recent years to investigate C_60_ has shown its prospective to mediate PDT of diverse diseases. Most of these reports have been limited to in vitro studies where not only cancer cells but also viruses, bacteria, and fungi [15,16] have been incubated with functionalized or solubilized C_60_ followed by light illumination. Light sources usually provide UV, blue, green or white because of the high C_60_ absorption in lower wavelengths with three intense bands in the UV region and a broad tail up to the red light [13,14]. Since in vivo PDT commonly uses red light for its tissue-penetrating properties, it was unclear whether C_60_ would mediate effective PDT in vivo. However, such concerns were addressed in a study of intraperitoneal photodynamic C_60_ therapy on a mouse model of abdominal dissemination of colon adenocarcinoma [50]. The synthesis of new C_60_ derivatives and nanocomplexes presents an alternative possibility to advance C_60_-based PDT [51,52].

Alternatively, rather than altering the photosensitizer molecule, research can also be focused on other sources of its excitation. Thus, deeper penetration of US waves into biological tissues provides an intriguing opportunity to use them as an alternative energy source for sensitizer excitation [29]. The present study evaluates perspectives of US in combination with pristine C_60_ as a sonosensitizer using a stable colloid pristine form for the treatment of carcinoma cells. US is used to deliver mechanical energy with its acoustic pressure wave in a non-invasive manner with minimal thermal effects due to its low intensity. Cavitation that occurs during acoustic pressure wave propagation through the liquid causes gas bubbles to implode with short bursts of light, known as sonoluminescence. Obtained results confirm the generation of sonoluminescence in the US bath and in the well of the plate exposed to ultrasound irradiation. The intensity of sonoluminescence increased with the output power of the US generator. The sonoluminescence spectrum [53] overlaps with the absorbance spectrum of C_60_, suggesting that it could induce the cytotoxic photosensitizing activity of C_60_ [14,54].

As C_60_ is able to penetrate lipid bilayers [55], it can translocate through the cell plasma membrane [7,18]. Russ et al. showed the role of endo-/phagocytosis in the cellular uptake of C_60_ is negligible [52]. Qiao et al. indicated by simulation that a pristine C_60_ molecule can rapidly pass the membrane [46]. The interaction of the C_60_ cluster with the membrane is followed by disaggregation of nC_60_ within the bilayer and diffusion of molecules through transient micropores [53]. C_60_ promotes the passive diffusion of the small molecules and induces endocytosis/pinocytosis preferentially in cancer compared to normal cells [54]. A low own fluorescence intensity challenged the direct investigation of C_60_ intracellular accumulation with simple and reliable fluorescence-based techniques. The development of a monoclonal antibody against C_60_ conjugated to bovine serum albumin [56] made the indirect immunostaining of pristine C_60_ molecules possible. Recently, we optimized this for human leukemic CCRF-CEM cells [14]. However, this technique could not be used to evaluate C_60_′s intracellular concentration and accumulation dynamics. In that case, the optimal solution would be using liquid chromatography mass-spectrometry analysis, which allows a definitive identification and reproducible quantification of trace-level analytes in complex samples. This method was previously reported to be an effective tool for C_60_ quantification in water samples [57] and CCRF-CEM cells [14,49]. The combination of those methods enabled visualization and quantification of the intracellular accumulation of pristine C_60_ in HeLa cells.

HeLa cells were shown to take up pristine C_60_ from the media in a time-dependent manner. The maximum intracellular C_60_ level reached 802 ± 66 ng/10^6^ cells after 24 h of incubation (Figure 3b). This is the same tendency we observed in our previous research with CCRF-CEM cells: The intracellular content of C_60_ fullerene reached its maximum of <250 ng/10^6^ CCRF-CEM cells after 24 h of incubation. A subsequent minor decrease of C_60_ fullerene content in leukemic cell extract at 48 h could be accounted for by its partial efflux from the cancer cells [14]. The co-staining with nuclear and mitochondrial markers pointed towards a mitochondrial localization, which was further confirmed with differential centrifugation and HPLC-ESI-MS analysis. C_60_ exhibited predominant localization within mitochondria, with 47% of its overall content in cell extract (Figure 3a). The mitochondrial localization could be linked with C_60′_s high electronegativity and a resulting affinity to the mitochondria-associated proton pool [7,58]. According to density functional theory simulations, C_60_ diffuses into the protonated mitochondrial intermembrane space, where it interacts with up to 6 protons, acquiring a positive charge [58]. A recent study [58] revealed that the antioxidant protective effect on *Escherichia coli* stems from C_60_-mediated proton transfer and intracellular interaction with free radicals. Hypothetically C_60′_s properties as a mitochondria-targeted agent [59] are based on similar mechanisms. This phenomenon is common in carboxy fullerenes [60] and other negatively charged carbon nanoparticles, such as single-walled carbon nanotubes [61].

A possible effect of pristine C_60_ aqueous colloid solution without US on HeLa cell viability was explored before a further investigation of its combinational effect with US. C_60_ in a concentration range from 0 to 40 µM had no significant effect on HeLa cell viability (≥93%) during treatment for 24 and 48 h. In addition, the evaluation of caspase 3/7 activity and plasma membrane phosphatidylserine translocation evidenced no effect of the treatment with 20 µM C_60_ on HeLa cells. HeLa cells, treated with C_60_, exhibited no caspase 3/7 activity increase and no phosphatidylserine translocation that pointed to the absence of cytotoxic and proapoptotic effect of C_60_ towards HeLa cells in the used concentrations.

In order to follow up on previous studies that have evidenced sonodynamic effects of the C_60_ derivatives towards cells in vitro [42,43,44], the US set-up for the treatment of cells was designed as a submersed model corresponding to a “well on water surface” configuration [55] (Figure 7). Constant monitoring of the possible US effects on the temperature of the liquids in well-plates as well as on the cell viability (including “solvent” controls with sterile water equal to the volume for C_60_) could confirm that the observed biological response can be attributed to the toxic effect of the combined treatment of cells with C_60_ and US. For investigation of the combined effect of C_60_ and US, HeLa cells were incubated in the absence or presence of 20 µM C_60_ for 24 h and exposed to 1 MHz US at the spatial average, temporal average intensity I_SATA_ in 5.4 W/cm^2^ for different exposure times (≤60 s). After another 48 h of incubation, their viability gradually decreased to 59 ± 5% (Figure 4), caspase 3/7 activity was induced (Figure 6a), and cell death differentiation analysis distinguished apoptosis in early and late stages under the action of sonodynamically excited C_60_ (Figure 6b,c). A similar tendency was observed by Nguyen et al. in HeLa cells after sonication with C_60_/PMPC complexes for 3 min with a sonication power and frequency output of 100 W and 42 ± 6 kHz. After sonication, the cell viability was decreased to 10% [45].

Caspase 3/7 activity was most strongly increased during 60 s US treatment in the presence of C_60_ as compared to other durations (Figure 6a), indicating a dose-dependent apoptosis induction during combined cellular treatment with C_60_ and 1 MHz US. Our results suggest the potential application of US in combination with pristine C_60_ for the sonodynamic treatment of cancer cells. Further optimization of the US treatment of cells and tests with a “sealed well” configuration [55,56] are planned to prevent any possible undesired US parameter variations in order to apply the combined treatment strategy with C_60_ and 1 MHz US to additional cancer models on cellular, tissue and animal levels. The exact mechanisms underlying the C_60_ sonoexcitation and apoptosis induction during SDT are yet unknown. Since the employment of fullerenes for cancer treatment is still at an early stage of development, close attention should be paid to an identification of possible biosafety and biodistribution of C_60_ formulation. Based on the expended data obtained on 2D and 3D cell culture in vitro the final C_60_ formulation and US exposure conditions could set a ground for the in vivo animal study. For therapeutic application, any possible side-effect of US on the body homeostasis should be excluded. However, the typical diagnostic imaging employs US in a very similar frequency range and is known to be safe [57]. As the WHO considers spatial and temporal average intensity I_SATA_ of US ≤ 3 W/cm^2^ as a safe limit for therapeutic ultrasound treatment [58], it may well be possible to adapt our present experimental model to a real therapeutic setting for treating diseases such as cancer.

## 4. Materials and Methods

### 4.1. Chemicals

Dulbecco’s modified Eagle’s medium (DMEM), phosphate-buffered saline (PBS), fetal bovine serum (FBS), penicillin/streptomycin, l-glutamine, and Trypsin were obtained from Biochrom (Berlin, Germany). Poly-D-lysine hydrobromide, Triton X100, Bovine Serum Albumin, p-phenylenediamine, glycerol and 3-(4,5-dimethylthiazol-2-yl)-2,5-diphenyl tetrazolium bromide (MTT) were obtained from Sigma-Aldrich Co. (St-Louis, MO, USA). Paraformaldehyde, toluene, 2-isopropanol, methanol and acetonitrile (both HPLC-MS grade), tris(hydroxymethyl)aminomethane and ethylene glycol-bis(β-aminoethyl ether)-N,N,N′,N′-tetraacetic acid, dimethylsulfoxide (DMSO) and trypan blue were used from Carl Roth GmbH + Co. KG (Karlsruhe, Germany).

### 4.2. C_60_ Synthesis

The pristine C_60_ aqueous colloid solution was prepared by C_60_ transfer from toluene to water using continuous ultrasound sonication as described by Ritter et al. [13]. The obtained aqueous colloid solution of C_60_ was characterized by 0.2 mM C_60_ concentration, 99% purity, stability, and homogeneity [13,48].

### 4.3. Matrix-Assisted Laser Desorption Ionization-Time of Flight Mass Spectrometry

An Axima Confidence Matrix Assisted Laser Desorption Ionization-Time of Flight Mass Spectrometry (MALDI-TOF-MS, Shimadzu, Kyoto, Japan) was used to determine the mass of molecular species in the C_60_ colloid solution. The sample (1 μL) was mixed with an equal volume of saturated matrix solution (6.5 mM 2,5-dihidrobenzoic acid in 0.1% trifluoroacetic acid, 50% acetonitrile) and spotted on a stainless steel target plate and dried. Desorption and ionization were achieved using a 337 nm nitrogen laser. Mass spectra were obtained at a maximal laser repetition rate of 50 Hz within a mass range from 0 to 3000 Da. The MALDI-TOF mass spectrometer was calibrated externally using a mixture of standard peptides: Bradykinin fragment 1–7 (757.40 Da), Angiotensin II (human, 1046.54 Da), P_14_R (synthetic peptide, 1533.86 Da) and ACTH fragment 18–39 (human, 2465.20 Da) from ProteoMass Peptide&Protein MALDI-MS Calibration Kit. In order to generate representative profiles, a total of 600 laser shots were accumulated and averaged for each sample. MALDI-TOF-MS data processing was performed using the LaunchpadTM v.2.9 Software (Shimadzu, Kyoto, Japan).

### 4.4. Dynamic Light Scattering

Short ultrasonication (30 s, 35 kHz) was applied to remove air bubbles. The size distribution of the C_60_ aqueous colloid solution was evaluated with a Zetasizer Nano S equipped with a He-Ne 633 nm laser (Malvern Instruments, UK). Data were recorded at 37 °C in backscattering mode at a scattering angle of 173°. C_60_, placed in disposable polystyrene cuvettes, was measured 15 times to establish average diameters and intensity distributions. The autocorrelation function of the scattered light intensity was analyzed by the Malvern Zetasizer Software (Malvern Instruments, UK) with the Smoluchowski approximation.

### 4.5. Ultrasound Exposure Set-Up

The water for the ultrasound water bath was previously degassed with the vacuum pump Savant UVS 400A SpeedVac (Thermo Fisher Scientific Inc., Berlin, Germany). For precise positioning of the plates inside the US water bath, especially the distance between transducer and plate, a plate holder was designed in SOLIDWorks (Dassault Systems, Waltham, MA, USA) and 3D printed by ViNN:Lab (Technical University of Applied Sciences Wildau, Germany). The position of the plate holder was aligned precisely with the US transducer and marked for identical positioning of the well plate during every experiment. The well plate, in that way, was positioned 25 mm from the US transducer. Plates with cells, seeded and treated with C_60_ according to the type of assay described below, were prepared for US treatment. In order to hinder overheating of the plate, every empty well, as well as the spaces between the wells on the plate, were filled with 100 μL of filtered water. The US treatment was performed with the US generator 68,101 coupled with an MH2 transducer, which was mounted on a water bath (Kaijo, Tokyo, Japan). The US transducer itself was a stainless steel transducer plate installed into a polypropylene tank filled with degassed water. The US transducer had an area of 136 × 81 mm and a frequency of 950 kHz (∼1 MHz). The apparatus for the US exposure is shown schematically in Figure 7. The US transducer was driven at 500 W in continuous mode, that correlated to the spatial average, temporal average intensity I_SATA_ of US in 5.4 W/cm^2^. The temperature of the sample solution was monitored with a digital thermometer. Thus, different locations of a well as well as a space between wells were compared during different US treatment duration. No temperature increase was found for the well plate filled with cell culture medium, preincubated at 37 °C and subjected to the US treatment for 60 s at 500 W, for which longer treatment duration a temperature increase was detected. Therefore, the ultrasound treatment duration was limited to 60 s.

### 4.6. Sonoluminescence Detection

Sonoluminescence measurements were performed directly in the US treatment set-up described before (Figure 7). The additional experimental set-up for sonoluminescence detection consisted of the Hacac (Hamamatsu Photonics, Japan), connected with the Oscilloscope Voltcraft 6150c (Conrad Electronic, Germany) and the power supply Thorn EMI PM28B (Thorn Lighting Ltd., United Kingdom). The 24-well plate was used because its wells match the diameter of the photomultiplier window. US bath and plate were filled with degassed distilled water for better sonication and sonoluminescence intensity [59]. The plate was placed on the plate holder in the US bath. A polyfoam holder was used to position the photomultiplier tube on top of a well of the 24-well plate. The US bath was additionally coated with aluminum foil, and measurements were performed in a dark room to shield the photomultiplier tube from any external light. The photomultiplier tube was used to detect sonoluminescence. The obtained data are presented as an average peak-to-peak voltage for the entire waveform (Vpp) during 120 s that indexes a full voltage between positive and negative peaks of the detected waveform of voltage on the photomultiplier tube.

### 4.7. Cell Culture

The human cervix adenocarcinoma cell line HeLa (ACC 57) was kindly provided by Dr. Müller (Division of Gastroenterology, Infectiology and Rheumatology, Charité—Universitätsmedizin Berlin, Germany). Human embryo lung HEL 299 cells were obtained from Hölzel Diagnostika Handels GmbH (Köln, Germany).

Cells were maintained in DMEM, supplemented with 10% FBS, 1% penicillin/streptomycin and 2 mM glutamine and cultured in 25 cm^2^ flasks at 37 °C with 5% CO_2_ in a humidified incubator binder (Tuttlingen, Germany). Treatment with Trypsin (1:10 in PBS) was used to detach adherent cells. The number of viable cells was counted upon 0.1% trypan blue staining with a Roche Cedex XS analyzer (Basel, Switzerland).

### 4.8. Visualization of Intracellular C_60_ Accumulation

HeLa cells (10^5^/mL) were seeded in 6-well plates on glass coverslips, previously coated with poly-D-Lysine, and incubated for 24 h. Cells were treated with 20 µM C_60_ colloid solution for a further 24 h. C_60_ molecules inside cells were visualized with immunofluorescence staining (Figure 8) and fluorescence microscopy. The synthesis of monoclonal antibodies against C_60_ was described by Hendrickson et al. [60]. Different types of immunoassay staining were described in The Immunoassay Handbook [61].

Specific fluorescent dyes were used for co-visualization of subcellular compartments such as mitochondria and nuclei—MitoTracker Orange FM (Invitrogen Molecular Probes, Carlsbad, USA) and 4′,6-diamidine-2′-phenylindole dihydrochloride (DAPI, Sigma-Aldrich Co., St-Louis, USA), respectively. For staining of the mitochondria, cells were washed with PBS and stained with the MitoTracker Orange FM for 30 min at 37 °C. Then, cells were fixed with 4% paraformaldehyde for 15 min in the dark and permeabilized with 0.2% Triton X100 for 10 min at room temperature and washed again with PBS. Primary monoclonal antibody IgG against C_60_ (Santa Cruz Biotech Inc., Santa Cruz, USA) and polyclonal antibody against mouse IgG F7506 labeled with fluorescein isothiocyanate (FITC, Sigma-Aldrich Co., St-Louis, USA) were subsequently used [14]. Finally, the coverslips were rinsed with dH_2_O, incubated with nucleus staining antifade solution (0.6 µM DAPI, 90 mM p-phenylenediamine in glycerol/PBS) for 2 h in the dark and sealed with slides.

Fluorescence microscopy was performed with the Keyence Microscope BZ-9000 BIOREVO (Osaka, Japan) equipped with blue (for DAPI, λ_ex_ = 377 nm, λ_em_ = 447 nm), green (for FITC, λ_ex_ = 472 nm, λ_em_ = 520 nm) and red (for MitoTracker, λ_ex_ 543 nm, λ_em_ = 593 nm) filters with the acquisition Software Keyence BZ-II Viewer (Osaka, Japan). The merged images and single-cell fluorescence intensity profiles were processed with the Keyence BZ-II Analyzer Software (Osaka, Japan).

### 4.9. Quantification of Intracellular C_60_ Accumulation

To study the accumulation dynamics, we have extracted C_60_ from the cell homogenate as well as from the mitochondrial fraction and carried out high-performance liquid chromatography-electrospray ionization mass spectrometry (HPLC-ESI-MS, Shimadzu, Kyoto, Japan) analysis as previously established [49].

Briefly, HeLa cells (10^5^/mL) were seeded in 6-well plates from Sarstedt (Nümbrecht Germany). After 24 h, cells were incubated for 0–48 h in the presence of 20 µM C_60_. Cells were washed with PBS three times, harvested and frozen-thawed in distilled H_2_O three times and dried at 80 °C under reduced pressure. C_60_ was extracted to toluene/2-isopropanol (6:1, *v/v*) via 1 h sonication. After centrifugation (70 min, 20,000× *g*), the toluene layer was analyzed with HPLC-ESI-MS. Chromatographic separation of C_60_ was performed using the column Eclipse XDV-C8 (Agilent, Santa Clara, USA) under isocratic elution conditions with a mobile phase of toluene and methanol. Optimized chromatographic conditions and MS parameters were recently published [14].

The mitochondrial fraction, obtained according to [62], was used for extraction of C_60_ as described above, as well as for measurements of protein concentration [63] and succinate-reductase activity [64], used as a mitochondrial marker to testify enrichment and purity of the fraction.

### 4.10. Cell Viability

HeLa and HEL 299 cells (10^4^/well), cultured in 96-well cell culture plates from Sarstedt (Nümbrecht, Germany) for 24 h, were treated with the 1% FBS DMEM medium containing 20 µM C_60_ for 24 h and exposure to the 1 MHz US treatment. The control cells were treated without and with an equal volume of sterile water as a solvent of C_60_ colloid solution. Cell viability was determined with an MTT reduction assay [65] at 48 h after US treatment. Briefly, cells were incubated for 2 h at 37 °C in the presence of 0.5 mg/mL MTT. The diformazan crystals were dissolved in DMSO and determined at 570 nm with a microplate reader Tecan Infinite M200 Pro (Männedorf, Switzerland).

Cell viability assay was accompanied by the phase contrast microscopy analysis of cells under the study with the Keyence BZ-9000 BIOREVO (Osaka, Japan).

### 4.11. Caspase 3/7 Activity

HeLa cells were seeded into 96-well plates (10^4^ cells/well) and incubated for 24 h. The cells were treated with 20 µM C_60_ for 24 h and subjected to US treatment (0, 20, 40, and 60 s) as described above. The activity of caspases 3/7 was determined at 24 h after ultrasound exposure using the Promega Caspase-Glo^®^ 3/7 Activity assay kit (Madison, USA) according to the manufacturer’s instructions. Briefly, the plates were removed from the incubator and allowed to equilibrate to room temperature for 30 min. After treatment, an equal volume of Caspase-Glo 3/7 reagent containing luminogenic peptide substrate was added, followed by gentle mixing with a plate shaker at 300 rpm for 1 min. The plate was then incubated at room temperature for 2 h. The luminescence intensity of the products of the caspase 3/7 reaction was measured with the microplate reader Tecan Infinite M200 Pro (Männedorf, Switzerland).

### 4.12. Cell Death Type Differentiation

HeLa cells, seeded in 6-well plates at a cell density of 6 × 10^4^ cells/well in 1.5 mL of culture medium, were incubated for 24 h, then the medium was replaced with a C_60_-containing medium. After 24 h of incubation with C_60,_ HeLa cells were treated with US, as indicated above. At 24 h after US treatment, cells were harvested. Apoptosis was detected by Annexin V-fluorescein isothiocyanate/propidium iodide apoptosis detection kit according to the manufacturer’s instructions. Briefly, cells were harvested and washed with binding buffer. After the addition of FITC-conjugated Annexin V, cells were incubated for 15 min at room temperature in the dark. Cells were washed with Binding buffer and, at 10 min after propidium iodide addition, were analyzed with the BD FACSJazz™ (BD Biosciences, Singapore). A minimum of 2 × 10^4^ cells per sample were acquired and analyzed with the BD FACS™ Software (BD Biosciences, Singapore).

On every histogram of flow cytometry four populations of cells are presented according to green (Annexin V-FITC) and red propidium iodide (PI) fluorescence intensities: viable (Annexin V-FITC negative, PI negative), early apoptotic (Annexin V-FITC positive, PI negative), late apoptotic (Annexin V-FITC positive, PI positive) and necrotic (Annexin V-FITC negative, PI positive) cells.

### 4.13. Statistics

All experiments were carried out with a minimum of four replicates. Data analysis was performed with the use of GraphPad Prism 7 (GraphPad Software Inc., San Diego, CA, USA). Paired Student’s *t*-tests were performed. The significance level was set at *p* < 0.01.

## Figures and Tables

**Figure 1 ijms-24-01020-f001:**
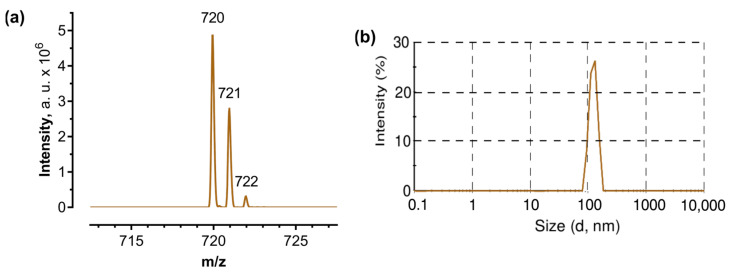
C_60_ aqueous colloid solution: (**a**)—MALDI-TOF-MS spectrum of C_60_ colloid solution, a.u. = arbitrary units; (**b**)—Hydrodynamic size (diameter, nm) of 20 µM C_60_, Intensity (%): percentage of all scattered light intensity.

**Figure 2 ijms-24-01020-f002:**
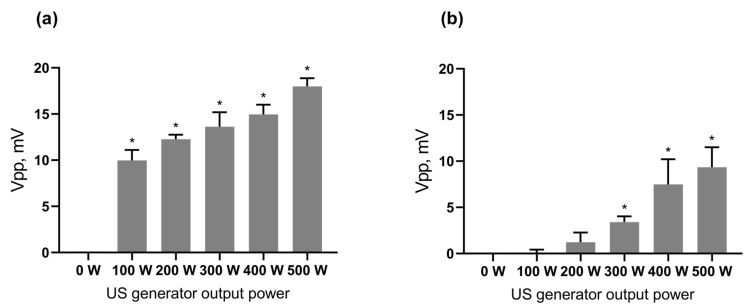
Sonoluminescence intensity: (**a**)—directly in the US bath, (**b**)—in the well of the plate, placed in the US bath; *—*p* ≤ 0.01 in comparison with 0 W output power.

**Figure 3 ijms-24-01020-f003:**
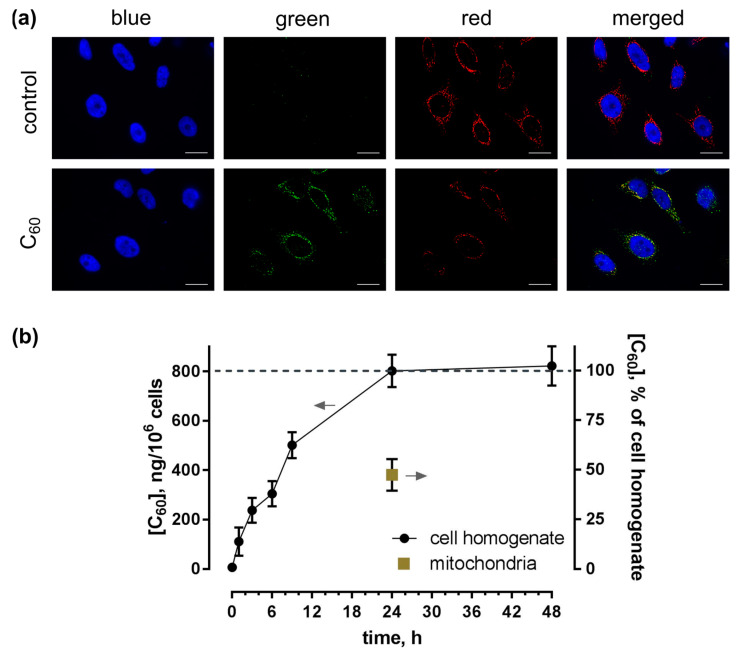
Uptake of C_60_ in HeLa cells: (**a**)—Fluorescence microscopy images of HeLa cells, incubated for 24 h with 20 µM C_60_ and stained with DAPI (blue), MitoTrecker (red) and FITC-labeled antibody against C_60_ (green), scale bar 20 µm; (**b**)—HPLC-ESI-MS analysis of C_60_ content in toluene extracts from cell homogenate and mitochondrial fraction after incubation of cells with 20 µM C_60_.

**Figure 4 ijms-24-01020-f004:**
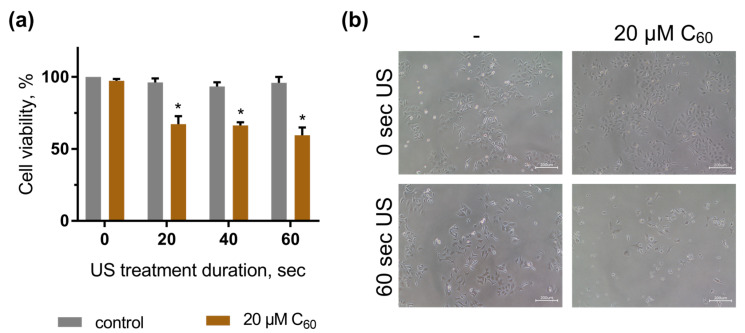
Viability of HeLa cells, incubated in the presence of 20 µM C_60_ and treated with 1 MHz ultrasound (US): (**a**)—MTT assay, *—*p* ≤ 0.01 in comparison with the viability of cells, treated with the respective duration of US; (**b**)—Phase contrast microscopy images of HeLa cells, incubated in the presence of 20 µM C_60_ and treated with 60-s ultrasound, scale bar 200 µm.

**Figure 5 ijms-24-01020-f005:**
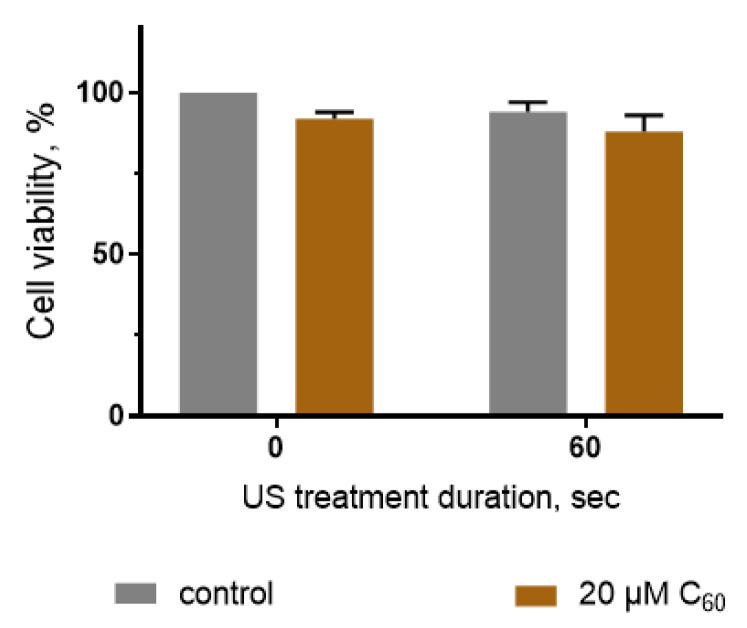
Viability of HEL 299 cells, incubated in the presence of 20 µM C_60_ and treated with 1 MHz ultrasound (US).

**Figure 6 ijms-24-01020-f006:**
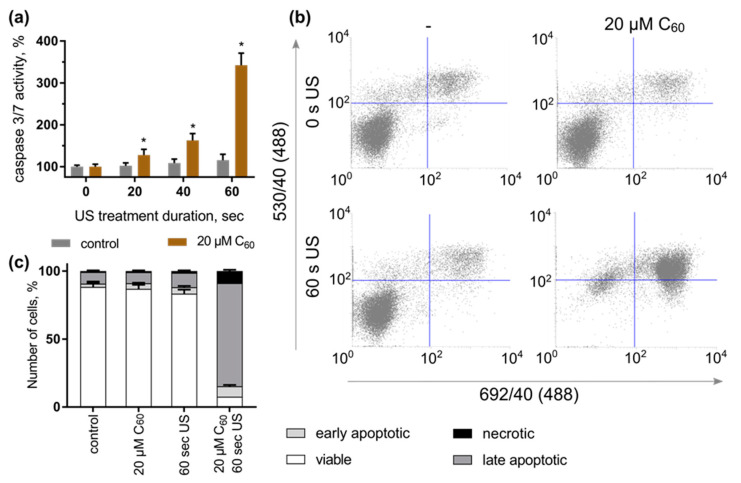
Apoptosis induction in HeLa cells by sonodynamically excited C_60_: (**a**)—Caspase 3/7 activity, *—*p* ≤ 0.01 in comparison with the viability of cells, treated with the respective duration of US; (**b**)—FACS histograms of HeLa cells, stained with Annexin V-FITC/PI (in each panel the lower left quadrant shows the content of viable, upper left quadrant—early apoptotic, upper right quadrant—late apoptotic, lower right quadrant—necrotic cells populations); (**c**)—Quantitative analysis of cell population content, differentiated with double Annexin V-FITC/PI staining.

**Figure 7 ijms-24-01020-f007:**
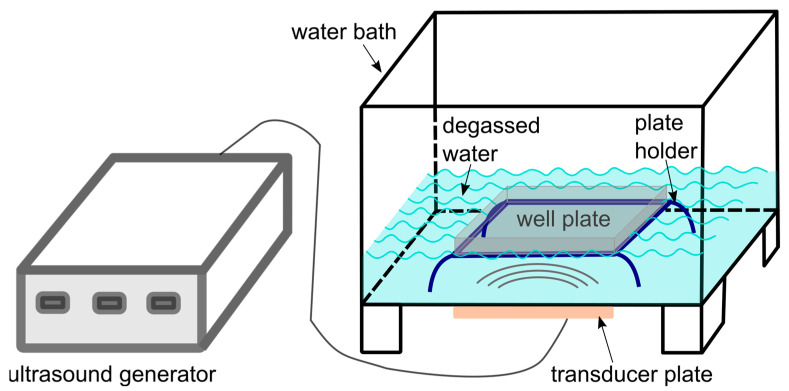
Diagram of the ultrasound exposure equipment.

**Figure 8 ijms-24-01020-f008:**
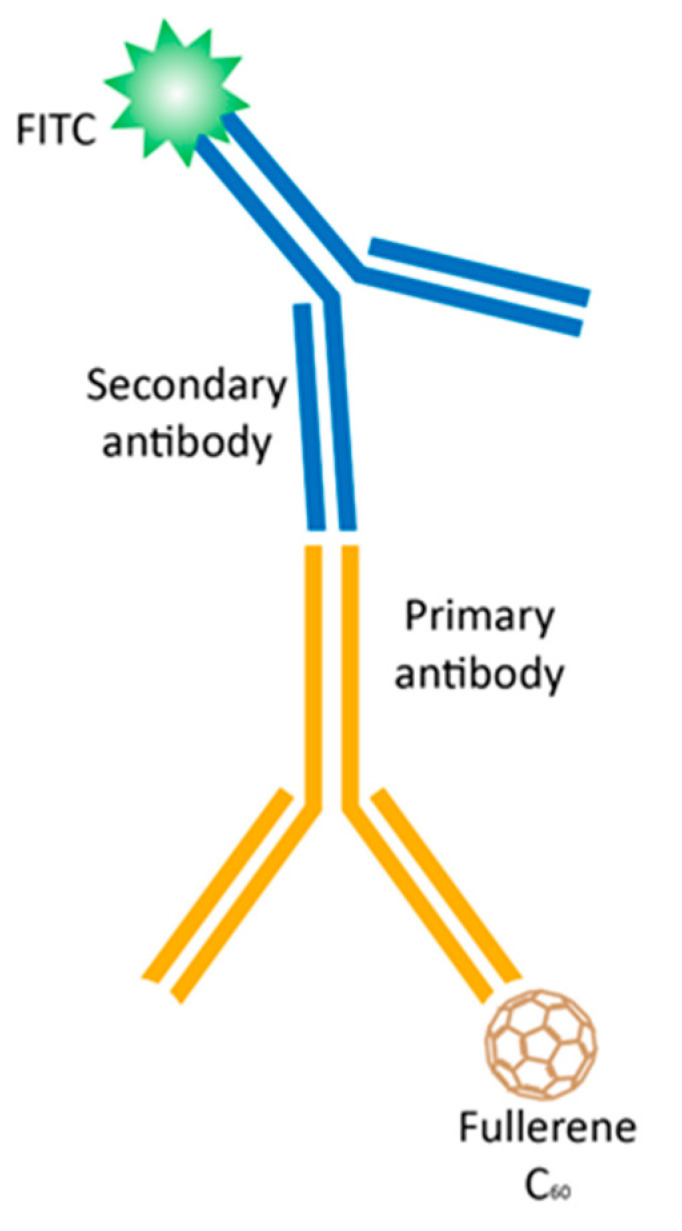
Immunofluorescence staining to assess the intracellular accumulation of C_60_: a primary monoclonal antibody IgG against C_60_ binds to C_60_; a FITC-labeled secondary antibody against the host species of the primary antibody binds to the primary antibody to allow detection with a fluorescence microscopy.

## Data Availability

The datasets used and analyzed during the current study are available from the corresponding author upon reasonable request.

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
