# Peer review of "Silent Death by Sound: C60 Fullerene Sonodynamic Treatment of Cancer Cells"

_ijms, 2023, doi:10.3390/ijms24021020_

Round 1
Reviewer 1 Report
In this manuscript, the authors described in vitro cell viability and the cell death mechanism analysis under sonodynamic treatments using aqueous solution of C60 cluster prepared in situ in sonicated water medium. There are serious questions regarding to the status of actual fullerene structures in such colloidal solution used in this study, with some comments as follows.
1. C60 molecule is highly hydrophobic with nearly zero solubility in water. The chemical structure of aqueous colloidal C60 cluster solution prepared under sonication conditions are extremely ambiguous and far from clear. Such colloid was suggested to be stabilized with the surface formation of H2O-compatible C60(OH)x without direct proof.
2. From (1), the question arises regarding what certain compounds or a number of complex compound mixtures being truly involved for active PDT or SDT in the study. Mechanistic sonodynamic effect interpretation of experimental results may be difficult and challenge to be correlated to the fullerene structures participated in vitro.
3. These C60 mixtures were extracted from either the cell homogenates or mitochondrial fractions, the authors should perform extend detailed HPLC-ESI-MS analyses on all C60 derivatives found other than C60 itself, as reported on page 5, since most biologically active fullerene derivatives are C60(OH)x analogues, not C60 itself, (example ref: Lu, L. H., et al., British J. Pharmacology 1998, 123, 1097.) Aggregation of highly hydrophobic C60 into solid particles are not bio-active in general, due to the fact that a large number of internal C60 molecules have no direct contact with O2 for PDT or SDT.
4. Regarding to the cell internalization of colloid C60, the measured size of 120 nm (Fig. 1b, page 3) is apparently too large for lipid bilayer transportation. Will the endocytosis mechanism be the path to be included in the discussion?
5. Internal localization at the mitochondrial region tends to be cationic compounds. Isolation of pristine C60 there should also indicate or partially provide evidence of a complex bulk nanoparticle composition of colloid fullerene structural mixtures containing hydrophilic cationic C60 derivatives at the particle surface.
6. Both C60 and C60(OH)x are detectable in MALDI-TOF spectrum. Authors confirmed the molecular ions of C60 at m/z 720, 721 and 722, without those ions for C60(OH)x. According to authors, MALDI-TOF peaks at m/z 722 and 723 were caused by 13C and 14C isotopes. However, these may be appropriate to be correlated to the molecular ion peaks of [M+H] and [M+2H] fragments in matrix or in the presence of TFA.
7. Perhaps, evaluation of the time dependent C60 uptake by HeLa cells showing internal localization can be accurately observed by microtome slides after cell fixation.
8. Regarding to the PDT or SDT effects with non-specific targeting, the same experiments should be carried out with the normal cells as the control for comparison.
Reviewer 3 Report
In this study, the effect of ultrasound activated C60 fullerene was evaluated. The study is interesting however authors should give more information about synthesized C60 fullerene such as mean particle size and standard deviation, PDI, zeta potential and shape of nanoparticles (e.g. electron microscopy images).
Round 2
Reviewer 1 Report
Author’s effort in finding many references as the response to the comments is very much appreciated. However, the revised manuscript itself did not resolve all the issues and questions in the original comments. Authors used many citations in the response to claim the C60 structure(s) without adding any actual experimental data in the revised manuscript to support the claim. Without full characterizations to prove the exact chemical structure(s) used for biotesting, the manuscript is not appropriate to be published in IJMS.
Reviewer 3 Report
The manuscript can be accepted in its current form.